# Regulation of microtubule dynamic instability by the carboxy-terminal tail of β-tubulin

Colby P Fees ⓘ, Jeffrey K Moore ⓘ

**Dynamic instability is an intrinsic property of microtubules; however, we do not understand what domains of αβ-tubulins regulate this activity or how these regulate microtubule networks in cells. Here, we define a role for the negatively charged carboxy-terminal tail (CTT) domain of β-tubulin in regulating dynamic instability. By combining in vitro studies with purified mammalian tubulin and in vivo studies with tubulin mutants in budding yeast, we demonstrate that β-tubulin CTT inhibits microtubule stability and regulates the structure and stability of microtubule plus ends. Tubulin that lacks β-tubulin CTT polymerizes faster and depolymerizes slower in vitro and forms microtubules that are more prone to catastrophe. The ends of these microtubules exhibit a more blunted morphology and rapidly switch to disassembly after tubulin depletion. In addition, we show that β-tubulin CTT is required for magnesium cations to promote depolymerization. We propose that β-tubulin CTT regulates the assembly of stable microtubule ends and provides a tunable mechanism to coordinate dynamic instability with ionic strength in the cell.**

## Introduction

Microtubules are polymers made of repeating subunits of αβ-tubulin heterodimers. Each heterodimer binds four neighboring heterodimers; longitudinal interactions form linear chains known as protofilaments and lateral interactions bind adjacent protofilaments [1]. In this way, heterodimers polymerize into a sheet known as the microtubule lattice, with curvature along the lateral axis that closes the lattice into a cylinder of 11–15 protofilaments with cytosolic and luminal surfaces. Polymerization is an intrinsic property of αβ-tubulin heterodimers, and the rate of polymerization depends on the concentration of available free tubulin. Microtubules also exhibit a unique nonequilibrium behavior of stochastically switching between polymerization and rapid depolymerization—a behavior known as dynamic instability. Like polymerization, dynamic instability is an intrinsic property of αβ-tubulin heterodimers. It is observed for microtubules assembled in vitro from purified αβ-heterodimers [2, 3, 4] and in living cells [5]. Cells use dynamic instability to organize networks of microtubules and to do work, such as segregating chromosomes during mitosis [6].

The conventional model for dynamic instability relies on allosteric coupling between αβ-tubulin heterodimers at the ends of microtubules, forming a stable cap. The stability of this cap is thought to depend on the nucleotide binding status of heterodimers near the microtubule end, with newly incorporated GTP-bound heterodimers promoting stability, hence the term "GTP cap" [2]. When the number of GTP-bound heterodimers at the microtubule end drops below a threshold level, the microtubule undergoes catastrophe and switches to rapid depolymerization. This model is supported by recent work showing a direct correlation between polymerization rate and the stability of the microtubule end [7]. An alternative model emphasizes the structure of the microtubule end as a key determinant of stability. As a microtubule grows, the asymmetric growth of some protofilaments causes the microtubule to form a tapered plus end with extensions of incomplete lattices. Tapered plus ends were originally observed by cryo-electron microscopic examination of microtubules assembled from purified tubulin and more recently by total internal reflection fluorescence (TIRF) microscopy examination of dynamic microtubules [8, 9]. Tapered plus ends tend to form in an age-dependent manner and under faster polymerization conditions, and are predicted to be less stable because of fewer lateral bonds between protofilaments [7, 8, 10]. Despite our growing understanding of dynamic instability, we do not understand how these basic mechanisms are regulated in cells to guide the formation and function of microtubule networks.

Several lines of evidence suggest that carboxy-terminal tail (CTT) domains of αβ-tubulin heterodimers might act as regulatory modules for controlling microtubule dynamics. CTTs are unstructured domains that extend off of the cytosolic surface of the microtubule lattice [11]. Early biochemical experiments showed that removal of CTTs by the nonspecific protease subtilisin enhances tubulin polymerization. This subtilisin-digested tubulin more readily assembled into large oligomers than undigested tubulin, as detected in bulk assembly assays, and these oligomers were more resistant to destabilization by calcium ions [12, 13, 14, 15]. However, the oligomers

Department of Cell and Developmental Biology, University of Colorado School of Medicine, Aurora, CO, USA

Correspondence: jeffrey.moore@ucdenver.edu

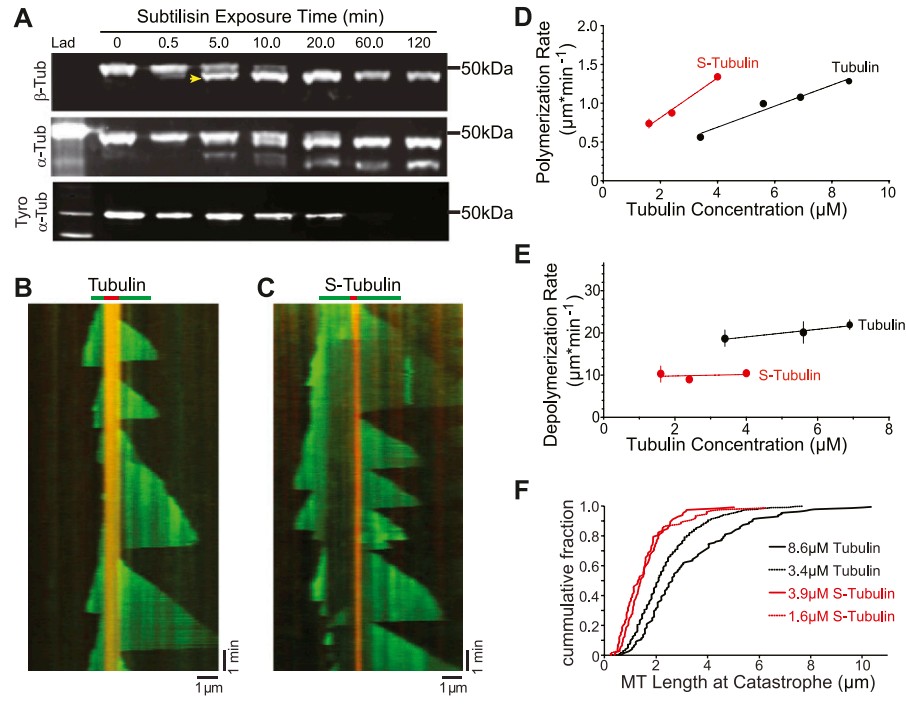

**Figure 1. β-CTT promotes microtubule dynamics.**
**(A)** Western blot of tubulin after a time course of subtilisin digestion. Porcine brain tubulin was treated with 1% subtilisin for indicated times at 30°C. Blots were probed with anti-β-tubulin (top), anti-α-tubulin (middle), and anti-α-tyrosinated tubulin (bottom). Arrowhead marks the faster migrating species of β-tubulin produced after 5-min digestion. **(B)** Representative kymograph of tubulin (green) polymerized from GMPCPP-stabilized microtubule seeds (red), collected at 3-s intervals. Tubulin concentration: 5.6 µM, vertical scale bar = 1 min, and horizontal scale bar = 1 µm. **(C)** Representative kymograph of S-tubulin as in (B). S-tubulin concentration: 4.9 µM, vertical scale bar = 1 min, and horizontal scale bar = 1 µm. **(D)** Polymerization rate plotted as a function of tubulin concentration. Data points are mean ± 95% CI plotted for each concentration. Each data point represents at least 197 polymerization events from at least 36 micro-tubules, collected from four separate experiments. Linear regressions were fit to the data and plotted as solid lines. Black: tubulin; red: S-tubulin. **(E)** Depolymerization rate plotted as a function of tubulin concentration. Each data point represents at least 62 depolymerization events from at least 28 microtubules, collected from three separate experiments. Data points are mean ± 95% CI plotted for each concentration. **(F)** Microtubule length at catastrophe plotted as the cumulative fraction of the total population. Tubulin concentrations: 3.4 (black dotted line) and 8.6 µM (solid black line); N = 502 and 125 catastrophe events, respectively. S-tubulin concentrations: 1.6 (red dotted line) and 3.9 µM (solid red line); N = 74 and 112, respectively. Source data are available for this figure.

formed by subtilisin-digested tubulin often exhibited aberrant and/or incomplete lattices ([12], [16], [17]). This evidence suggests that CTTs may negatively regulate tubulin assembly to guide the proper formation of the microtubule lattice. In the in vivo context, CTTs in different cell types exhibit genetically encoded and posttranslational differences. Whereas the globular domains of α- and β-tubulin are highly conserved, CTTs exhibit diverse amino acid sequences when compared across species and between tubulin isotypes within a species ([18]). In addition, CTTs are targeted for a variety of post-translational modifications ([19], [20]). This raises the possibility that CTTs may act as regulatory handles for changing the functional properties of microtubules; however, the roles of CTTs in regulating microtubule dynamics and interactions remain poorly defined.

We previously characterized the roles of α- and β-CTT during mitosis in budding yeast and identified an important role for β-CTT in promoting proper spindle formation and chromosome segre-gation ([18]). Live-cell imaging and electron tomography of mutant yeast cells with genetically ablated β-CTT showed disorganized spindle microtubules. Therefore, we hypothesized that β-CTT may be required for cells to properly regulate microtubule dynamics during spindle assembly.

Here, we test this hypothesis using a combination of in vitro experiments with purified mammalian tubulin digested with sub-tilisin and in vivo experiments in budding yeast with mutants that alter or ablate β-CTT. We show that β-CTT contributes to micro-tubule dynamics by inhibiting polymerization and promoting de-polymerization. Despite these inhibitory effects on microtubule assembly, we found that microtubules with β-CTT are less prone to catastrophe and exhibit distinct plus-end morphologies. Fi-nally, we show that the ability of magnesium ions to accelerate

microtubule depolymerization requires β-CTT. Together, our findings define a role for β-CTT in regulating dynamic instability and suggest a mechanism for regulating microtubule function and organization through ionic control.

# Results

## β-CTT promotes microtubule dynamics

To investigate how β-CTT impacts microtubule dynamics, we used a well-established protocol for proteolytically removing tubulin CTTs using the nonspecific protease subtilisin. Previous work has shown that subtilisin removes β-CTT from purified tubulin before removing the α-CTT ([14]). Consistent with this, we found that limited proteolysis of purified porcine brain tubulin with subtilisin pref-erentially removes β-CTT. In our experiment, 5-min digestion with subtilisin shifts the mobility of β-tubulin in SDS–PAGE, with ap-proximately 70% of total β-tubulin protein running at a slightly lower molecular weight (Fig 1A, arrowhead, and Fig S1). We used mass spectrometry to confirm that this sample contains β-tubulin species that are truncated at their carboxy termini (Fig S2 and Table S1). In contrast, removing α-CTT requires a longer subtilisin digest. Although our mass spectrometry analysis did identify some trun-cated species of α-tubulin after 5-min digest, Western blot analysis indicates that only 20% of total α-tubulin exhibits a mobility shift and the amount of tyrosinated α-tubulin is minimally affected (Figs 1A, S1, and S2, and Table S1). Thus, 5-min digestion with subtilisin produced a tubulin sample that predominantly lacks β-CTT. We refer to this sample as S-tubulin.

We first compared the polymerization and depolymerization rates of S-tubulin to untreated tubulin using TIRF microscopy to image individual microtubules grown from guanosine-5'-[(α,β)-methyleno]triphosphate (GMPCPP)–stabilized seeds (Fig 1B and C). By testing a range of concentrations, we found that S-tubulin exhibits an apparent on-rate constant of 7.4 subunits·μM$^{-1}$·s$^{-1}$, which is nearly twofold greater than the apparent on-rate constant of 3.96 subunits·μM$^{-1}$·s$^{-1}$ that we measured for untreated porcine brain tubulin (Fig 1D). Microtubules assembled from S-tubulin depolymerize significantly slower (median = 250 subunits·s$^{-1}$; 95% confidence interval [CI] = 245–258) than those assembled from untreated tubulin (median = 537 subunits·s$^{-1}$; 95% CI = 516–560; Fig 1E). Together, these findings indicate that β-CTT promotes dynamic microtubules by inhibiting polymerization and accelerating depolymerization.

Our TIRF experiments also permit analysis of catastrophe events, which are not accessible by bulk assays. We found that microtubules assembled from S-tubulin catastrophe at shorter lengths than microtubules assembled from untreated tubulin (Fig 1F). This difference is most striking when comparing concentrations of S-tubulin and untreated tubulin that polymerize at similar rates (Fig 1D and F, compare 3.9 μM S-tubulin and 8.6 μM untreated tubulin). This suggests that β-CTT prevents catastrophes.

In addition to the effects of β-CTT on the behavior of microtubule plus ends, we also observed differences in the behavior of minus ends. Polymerization from both ends of the GMPCPP seeds is more common in the presence of S-tubulin than untreated tubulin (Fig 1B and C). However, there is still a clear asymmetry between the two microtubule ends; polymerization at one end is always slower and reaches shorter terminal lengths. We presume that the slower end is the minus end. Thus, β-CTT appears to impact microtubule assembly at both plus and minus ends.

### β-CTT alters the structure of the plus end

Our finding that microtubules catastrophe more often in the absence of β-CTT, together with the previous finding that S-tubulin forms aberrant lattices in unseeded assembly assays, prompted us to examine whether β-CTT alters the morphology of growing microtubule plus ends (12, 17). Normally, plus-end morphology correlates with polymerization rate; microtubules polymerizing at slower rates exhibit blunt ends, whereas microtubules polymerizing at faster rates exhibit plus ends with long extensions of incomplete lattices, also known as tapered ends (7, 8, 10). We predicted that removing β-CTT could promote faster polymerization and more frequent catastrophes either by supporting longer plus-end tapers or by maintaining blunt ends at faster polymerization rates. We tested this using fluorescence analysis to measure the steepness of the signal decay at the plus ends of growing microtubules in our TIRF experiments (Fig 2A). In our analysis, a blunt plus end has a steeper fluorescence decay, whereas a tapered end has a more gradual decay. The length of the tapered end can be estimated by fitting the fluorescence decay to a Gaussian survival curve and calculating the SD (i.e., tip SD), as previously described (8).

Consistent with previous studies, we found that control microtubules assembled from untreated tubulin exhibit plus-end tapers and that these extensions are longer for microtubules polymerizing at faster rates (Fig 2A, C, and D) (8). In contrast, microtubules assembled from S-tubulin exhibit less tapered and more blunted plus ends, even at faster rates of polymerization (Fig 2B–D). These results indicate that β-CTT determines how heterodimers assemble at the plus end and promotes the formation of plus-end tapers.

### β-CTT promotes plus-end stability

Having found that β-CTT regulates plus-end morphology, we next examined its effects on plus-end stability. We used a technique modified from Walker et al (4) to measure plus-end behavior in the absence of polymerization. In our experiment, microtubules polymerize from GMPCPP seeds affixed to the coverslip for approximately 30 s. Then, free tubulin is washed out to prevent further polymerization by pipetting 4–5× chamber volumes of assembly buffer through the imaging chamber while continuously acquiring images (Fig 3A). Images are acquired at 1-s time intervals to detect rapid changes in microtubule length. After washout, the microtubules exhibit a period of slow depolymerization (Fig 3B, labeled "S") before undergoing catastrophe and switching to faster depolymerization (Fig 3B, labeled "F"). We computationally determined the time point of catastrophe by iteratively fitting microtubule lengths across five consecutive time

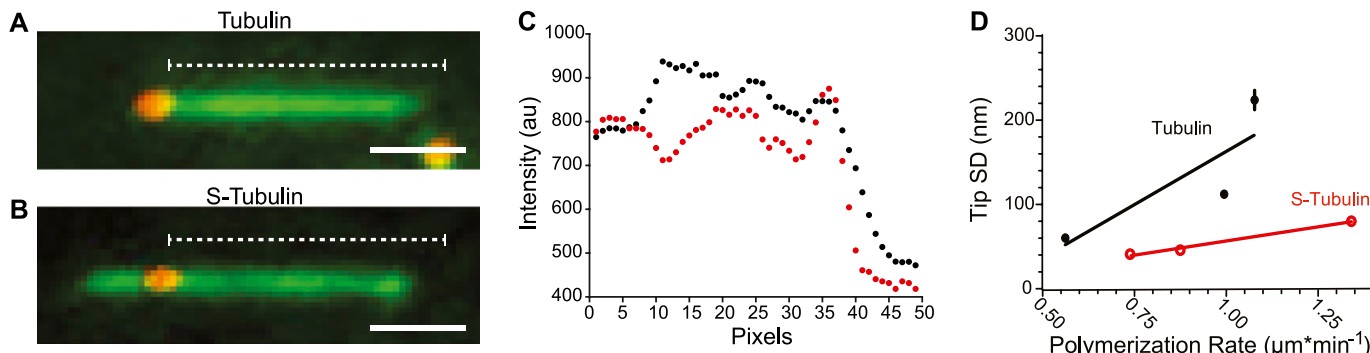

**Figure 2. β-CTT alters the structure of the microtubule plus end.**
**(A)** Representative image of a microtubule assembled from the untreated tubulin (6.9 μM). Dashed white line is the coverage of the line scan across the microtubule with intensities plotted in (C). Scale bar = 1 μm. **(B)** Same as (A) assembled with S-tubulin (4.0 μM). **(C)** Intensity measurements plotted as a function of pixel position of the microtubules in (A) (black) and (B) (red). **(D)** Mean tip SD plotted as a function of polymerization rate for tubulin (black) and S-tubulin (red). N = 355 time points per polymerization rate, from at least five different microtubules. Data points are mean ± 95% CI plotted for each concentration.

points to a linear regression. Catastrophe was defined as the first time point when the slope of the regression was larger than 150 nm·s$^{-1}$ (~2 pixels·s$^{-1}$; see the Materials and Methods section). We calculated time to catastrophe as the number of seconds between washout initiation and catastrophe (Fig 3B, labeled "D"). Under these conditions, the time between washout and catastrophe represents a readout of the stability of the plus end (4, 21).

We compared the time to catastrophe of S-tubulin microtubules to undigested controls in the same polymerization rate range using S-tubulin and untreated tubulin at concentrations that support a similar range of polymerization rates (2.3–4.6 μM and 3.1–11.6 μM, respectively). We found that time to catastrophe is significantly shorter for S-tubulin microtubules, with most (55%) exhibiting no detectable delay between washout and catastrophe (Fig 3C). The other 45% of S-tubulin microtubules do exhibit a delay before catastrophe; however, this time is significantly reduced compared with that for controls matched for polymerization rates (Fig 3D). Based on these findings and our results in Fig 1, we conclude that the plus ends of microtubules assembled from S-tubulin are more prone to catastrophe.

## β-CTT promotes dissociation from microtubules

Our washout experiments also allowed us to estimate rates of tubulin dissociation from microtubules in conditions that are not confounded by high concentrations of free tubulin. We first measured the slow rate of depolymerization that occurs before catastrophe (Fig 3B, labeled "S"). This slow depolymerization is thought to represent the dissociation of GTP-bound heterodimers from the plus end and is different from the faster depolymerization that follows catastrophe (21). Importantly, this dissociation rate can be used to determine the affinity of free tubulin for the polymer.

If β-CTT acts to repel heterodimers from the microtubule, we predicted that removing β-CTT would decrease the dissociation rate. Consistent with our model, we observed a slight but significant reduction in dissociation rates for the S-tubulin microtubules that did not immediately catastrophe after washout (S-tubulin median = 28 0.14 subunits·s$^{-1}$, 95% CI = 22–40; untreated control median = 35 subunits·s$^{-1}$, 95% CI = 32–43; Fig 3E). This suggests that S-tubulin heterodimers have a higher affinity for microtubule polymer than undigested tubulin.

We also measured the fast rate of depolymerization after catastrophe, defined by the slope of a linear regression of the microtubule lengths after catastrophe (Fig 3B, labeled "F"). S-tubulin exhibits a significantly slower depolymerization rate than untreated control, consistent with measurements from our earlier dynamics experiments (Figs 1E and 3F). Therefore, microtubules assembled from S-tubulin depolymerize slower than undigested controls, and this difference is independent of free tubulin. Taken together, these results support our conclusion that β-CTT regulates the equilibrium between microtubule

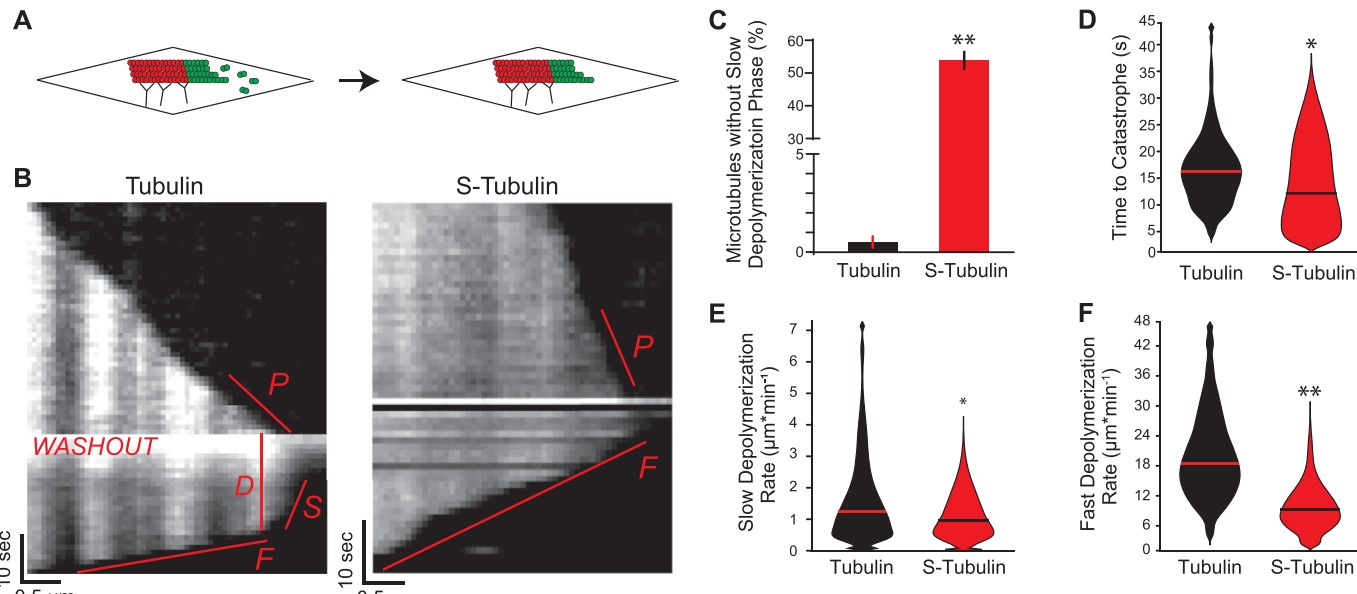

**Figure 3. β-CTT promotes microtubule plus-end stability.**
**(A)** Schematic of the washout experiment. **(B)** Representative kymographs of untreated tubulin (6.3 μM) and S-tubulin (4.3 μM) in the washout experiment. Labels denote aspects analyzed for the washout experiment, including P—polymerization rate, D—delay time to catastrophe, S—slow depolymerization rate, and F—fast depolymerization rate. S-tubulin kymograph does not exhibit the D and S states. **(C)** Percentage of microtubules without a slow depolymerization rate (B, labeled "S") for all washout experiments (number of microtubules without S state/total number of microtubules × 100). N = 180 tubulin and n = 94 S-tubulin microtubules, pooled from at least four different experiments for each. Error bars are standard error of the proportion. **P ≪ 0.001, significance determined by Fisher's exact test. **(D)** Time to catastrophe (B, labeled "D") in seconds for tubulin and S-tubulin microtubules after washout. N = 179 tubulin and n = 51 S-tubulin microtubules, pooled from at least four different experiments. Lines represent the median. Microtubules without a detectable slow depolymerization phase were excluded from this analysis. *P = 0.04, significance determined by the Mann–Whitney U test. **(E)** Slow depolymerization rate (B, labeled "S") for tubulin and S-tubulin microtubules after washout. N = 162 tubulin and n = 41 S-tubulin microtubules, pooled from at least four different experiments. Lines represent the median. Microtubules without a detectable slow depolymerization phase and/or a rate calculated to be zero were excluded from this analysis. *P = 0.05, significance determined by the Mann–Whitney U test. **(F)** Fast depolymerization rate (B, labeled "F") for tubulin and S-tubulin microtubules after washout. N = 179 tubulin and n = 92 S-tubulin microtubules, pooled from at least four different experiments for each. **P ≪ 0.001, significance determined by the Mann–Whitney U test.

polymers and free heterodimers; removing β-CTT shifts this equilibrium and stabilizes the polymer state.

## Magnesium cations regulate tubulin equilibrium through β-CTT

How might β-CTT promote the dissociation of tubulin subunits from microtubules? We hypothesized that the negatively charged β-CTT might facilitate the destabilizing effects of positively charged divalent cations, such as calcium and magnesium. For our experiments, we studied the effects of magnesium cations, which are not only important for GTP-binding and tubulin polymerization but also promote depolymerization (4, 21). We first tested how magnesium impacts microtubule dynamics by decreasing the concentration of $MgCl_2$ in our assembly buffer from 5 to 1 mM. As expected, lower magnesium decreases the rate of polymerization for both S-tubulin and untreated tubulin (Fig 4A). However, we observed different effects on the rate of depolymerization. Whereas the depolymerization rate of S-tubulin is unaffected by decreasing $MgCl_2$ from 5 to 1 mM, the depolymerization rate of untreated tubulin is significantly slower at lower $MgCl_2$ concentrations (Fig 4B).

To further test our hypothesis and isolate the effect of divalent cations on microtubule depolymerization, we modified our washout experiment. We first assembled microtubules from GMPCPP-stabilized seeds in the presence of buffer with 5 mM $MgCl_2$ using tubulin concentrations selected to achieve similar polymerization rates between S-tubulin and untreated tubulin (4.8–6.8 μM and 7.8–12.0 μM, respectively). We then washed out the free tubulin with buffer containing a fivefold lower concentration of magnesium (1 mM). Therefore, the microtubules analyzed in this experiment

were assembled in the presence of 5 mM $MgCl_2$ but depolymerize in the presence of 1 mM $MgCl_2$. We compared the rates of slow and fast depolymerization (i.e., before and after catastrophe, respectively) with our washout experiments in Fig 3, which have a constant concentration of 5 mM $MgCl_2$. Consistent with previous results, we found that magnesium strongly affects the depolymerization of microtubules assembled from untreated tubulin. Both the rates of slow and fast depolymerization are significantly decreased in the presence of lower magnesium concentration (Fig 4C and D). In contrast, microtubules assembled from S-tubulin are less sensitive to the change in magnesium concentration. We found no difference in the rate of slow depolymerization and only a slight decrease in the rate of fast depolymerization in the presence of low magnesium for S-tubulin (Fig 4C and D). These findings indicate that β-CTT is necessary for magnesium to promote tubulin dissociation and destabilize microtubules.

We next sought to test whether β-CTT mediates the effects of magnesium on microtubule stability in living cells. We turned to budding yeast as a model system, which affords two key advantages for our study. First, in contrast to many other eukaryotes, budding yeast express a single β-tubulin gene, *TUB2*. Second, *TUB2* coding sequence can be readily altered by genetic mutations at the chromosomal locus, allowing precise manipulation of the amino acid composition of β-CTT. We used this strategy to create a mutant yeast strain that lacks the final 27 amino acids of β-tubulin but retains helix 12 (*tub2-430Δ*) (22).

We predicted that β-CTT might regulate microtubule dynamics in response to shifts in divalent cation concentrations in cells. Accordingly, β-CTT may become essential for processes that require

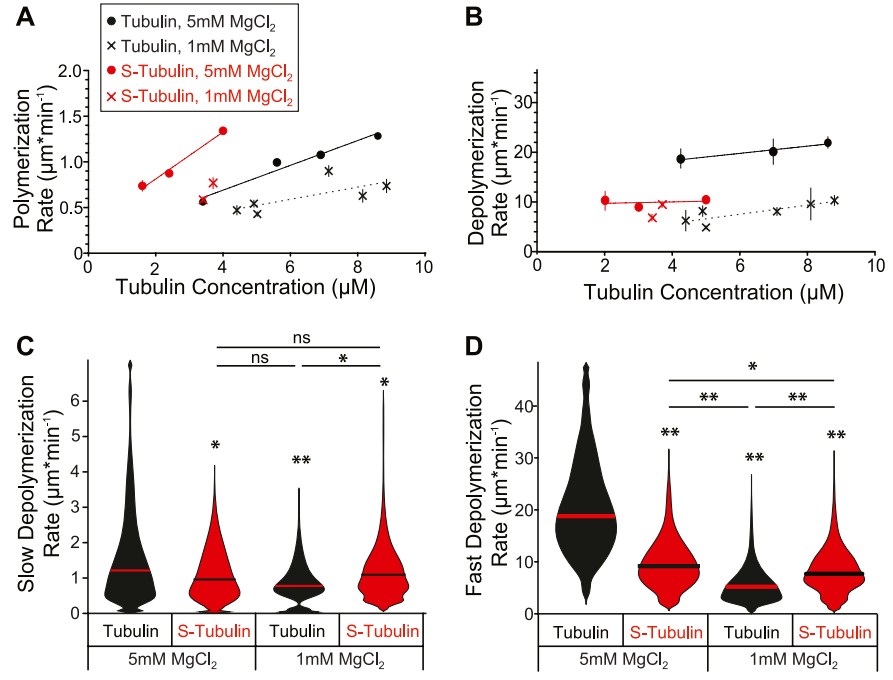

**Figure 4. Magnesium regulates tubulin equilibrium through β-CTT.**
**(A)** Polymerization rate plotted as a function of tubulin concentration. Crosses represent data from experiments conducted at 1 mM $MgCl_2$ and dots are data points from Fig 1D from experiments conducted at 5 mM $MgCl_2$. Black: tubulin and red: S-tubulin. Data points are mean ± 95% CI plotted for each concentration. Each data point represents at least 61 polymerization events from at least 24 microtubules, collected from at least three separate experiments. Linear regressions were fit to the data with at least three different concentrations and plotted as lines. **(B)** Depolymerization rate plotted as a function of tubulin concentration. Each data point represents at least 45 depolymerization events from at least 24 microtubules, collected from at least three separate experiments. Linear regressions were fit to the data with at least three different concentrations and plotted as lines. **(C)** Slow depolymerization rate (Fig 3B, labeled "S") for tubulin and S-tubulin microtubules after washing out free tubulin. The $MgCl_2$ concentration indicated is for the washout buffer. The 5 mM plots are data from Fig 3E, provided for comparison. Data for the 1 mM $MgCl_2$ represent 45 microtubules for untreated tubulin (4.8–6.8 μM) and 158 microtubules for S-tubulin (7.8–12.0 μM), pooled from at least four different experiments for each. Microtubules without a detectable slow depolymerization rate and/or a rate calculated to be zero were excluded from this analysis. *$P = 0.05$ and **$P \ll 0.001$, significance determined by the Mann–Whitney $U$ test. **(D)** Fast depolymerization rate (Fig 3B, labeled "F") for tubulin and S-tubulin microtubules after washing out free tubulin. The 5-mM plots are data from Fig 3F, provided for comparison. Data for the 1 mM $MgCl_2$ represent 52 microtubules for untreated tubulin (4.8–6.8 μM) and 260 microtubules for S-tubulin (7.8–12.0 μM), pooled from at least four different experiments for each. N = 179 tubulin and n = 92 S-tubulin microtubules, pooled from at least four different experiments for each. *$P = 0.01$ and **$P \ll 0.001$, significance determined by the Mann–Whitney $U$ test.

dynamic microtubules when divalent cations are depleted. In support of this notion, our previous genome-wide genetic interaction screen identified a negative interaction between the *tub2-430Δ* mutant and *alr2Δ*, a null mutation that disrupts one of the magnesium transporters at the yeast plasma membrane (23, 24).

To further test whether loss of β-CTT sensitizes cells to low magnesium, we used growth assays to compare fitness during magnesium depletion. Wild-type cells exhibit robust growth in standard synthetic media, which contains 4 mM MgSO$_4$, but are severely inhibited in media without MgSO$_4$ (Fig 5A; see the Materials and Methods section). Adding back 10 μM MgCl$_2$ restores growth to intermediate levels, whereas adding back 50 μM MgCl$_2$ fully restores growth (Fig 5A). As a positive control for these experiments, we tested the *mnr2Δ* null mutant, which disrupts access to magnesium stores in the vacuole (25). *mnr2Δ* mutant cells are impaired at 10 μM MgCl$_2$ but exhibit improved growth at 50 μM MgCl$_2$ (Fig 5B). The *tub2-430Δ* mutant that lacks β-CTT is impaired at 10 μM MgCl$_2$ and at 50 μM MgCl$_2$ (Fig 5B). Therefore, β-CTT is important for cell proliferation under low-magnesium conditions.

We next asked whether β-CTT destabilizes microtubules in vivo and whether this role depends on intracellular magnesium levels. We measured dynamic astral microtubules in preanaphase cells, which exhibit individual microtubules emanating from the two spindle poles (Fig 5C). We collected time-lapse image series at 4- to 5-s intervals, measured the lengths of the astral microtubules at each time point for at least 5 min and then compared the distributions of astral microtubule lengths for different magnesium conditions (26). We found that depleting magnesium by shifting

*mnr2Δ* mutants to media without MgSO$_4$ is sufficient to significantly increase microtubule length in cells expressing wild-type tubulin, when compared with controls having normal magnesium levels (*P* < 0.001; Fig 5C and D). To determine if this magnesium sensitivity depends on β-CTT, we examined *tub2-430Δ* mutants under normal and magnesium-depleted conditions. We found that microtubules are significantly longer in *tub2-430Δ* mutants, whether in normal or magnesium-depleted conditions (Fig 5C and D).

To confirm that the changes in astral microtubule length are not caused by indirect effects of magnesium depletion on cell cycle progression, we measured the lengths of astral microtubules during the G1 phase of the cell cycle (Fig 5E). During G1, long astral microtubules emanate from the single spindle pole, whereas nuclear microtubules are extremely short and keep kinetochores anchored proximal to the pole (27). The combined lengths of the astral microtubules in a G1 cell provide an estimate for the amount of microtubule polymer in that cell; however, the caveat of this experiment is that astral microtubules often form bundles during G1 that can be difficult to distinguish by light microscopy. Nevertheless, we observed trends that are similar to those in preanaphase cells. *tub2-430Δ* mutants that lack β-CTT exhibit significantly more microtubule polymer per cell than wild-type controls, and double mutants combining *mnr2Δ* with *tub2-430Δ* that are shifted to media without MgSO$_4$ exhibit a similar amount of total microtubule polymer compared with *tub2-430Δ* single mutants grown in the presence of MgSO$_4$ (Fig 5E and F). We conclude that low intracellular magnesium stabilizes microtubules, high intracellular magnesium destabilizes microtubules, and this relationship depends on β-CTT. However,

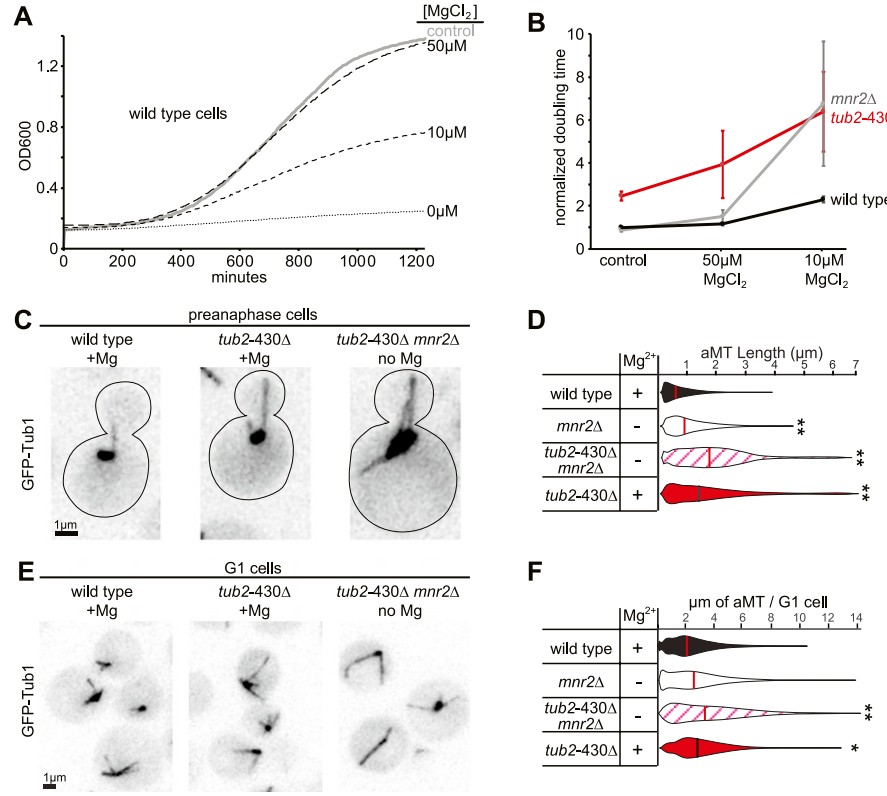

**Figure 5.  β-CTT regulates magnesium sensitivity in vivo.**
**(A)** Representative growth curves of wild-type cells in different magnesium conditions. Control group was cultured in synthetic media with 4 mM MgSO$_4$. Other groups were grown in synthetic media without MgSO$_4$ but supplemented with MgCl$_2$ at concentrations indicated. All cultures were grown at 30°C with agitation for 21 h, and OD600 was measured every 5 min. **(B)** Median doubling times of wild-type, *mnr2Δ*, and *tub2-430Δ* cells at indicated magnesium conditions, normalized to the doubling time of wild-type cells in 4 mM MgSO$_4$. Each data point represents at least 11 replicates from four separate experiments. Error bars are ± 95% CI. **(C)** Representative images of cells in preanaphase expressing GFP-labeled microtubules, grown in synthetic media or without MgSO$_4$. Scale bar = 1 μm. **(D)** Distribution of astral microtubule (aMT) lengths measured in preanaphase cells from an asynchronous culture. Lengths were measured every 4 to 5 s for 5 min. Cells were cultured in synthetic media with (+) or without (–) 4 mM MgSO$_4$. Data pooled from two separate experiments, with at least seven cells and 396 total measurements for each group. \*\**P* ≪ 0.001. Significance determined by the Mann–Whitney *U* test. Lines denote median. **(E)** Representative images of cells in G1 phase expressing GFP-labeled microtubules, grown in synthetic media or without MgSO$_4$. Scale bar = 1 μm. **(F)** Distribution of aMT lengths measured in G1 from an asynchronous culture. Cells were cultured in synthetic media with (+) or without (–) 4 mM MgSO$_4$. Data pooled from two separate experiments, with at least 60 cells for each group. \**P* = 0.01, \*\**P* ≪ 0.001. Significance determined by the Mann–Whitney *U* test. Lines denote median.

removing β-CTT alone appears to have a stronger stabilizing effect than depleting magnesium.

## The negative charge of β-CTT regulates tubulin equilibrium in vivo

We reasoned that the role of the β-CTT in regulating the tubulin equilibrium might depend on its enrichment for negatively charged amino acids. To test this hypothesis, we created a series of mutant yeast strains that either remove amino acids from β-CTT, substitute negatively charged side chains for neutral side chains, replace the native yeast tail with sequences that mimic tails from human β-tubulins, or swap the tail sequences from α- and β-tubulin (Fig 6A).

We applied our panel of β-CTT mutants in two experiments to test the prediction that negatively charged amino acids in β-CTT inhibit the microtubule polymer state. First, we destabilized microtubules by treating log-phase cultures of cells with the microtubule poison nocodazole and then fixed and imaged the cells to assess the degree of microtubule loss. Individual cells were scored for the presence or absence of astral microtubules (Fig 6B). Under these conditions, we found that approximately half of wild-type cells lack astral microtubules after 1 h in 1.5 μM nocodazole (Fig 6C). In contrast, mutants that lack β-CTT (tub2-430Δ) are resistant to nocodazole and most cells maintain astral microtubules (Fig 6A–C). This result is consistent with our in vitro work and suggests that removing β-CTT shifts the tubulin equilibrium toward the polymer state.

We next tested the relevance of negatively charged amino acids in β-CTT. We found that adding back a minimal region that is enriched for negatively charged amino acids restores sensitivity to nocodazole, resulting in microtubule loss that is similar to wild-type controls (tub2-438Δ; Fig 6A and C). We previously defined this domain as the "acidic patch" because it contains the greatest enrichment of negatively charged residues (Fig 6A) (18). To confirm the importance of the negatively charged side chains in the acidic patch, we mutated glutamate and aspartate residues to glutamine, aiming to preserve the structure of the residues while neutralizing the charge. Mutating the acidic patch alone (tub2-polyQ) or combing the neutralizing mutations with a truncation of the rest of β-CTT (tub2-polyQ-438Δ) is sufficient to confer resistance to depolymerization by nocodazole (Fig 6A and C).

As a complementary approach to better understand the role the acidic patch plays in regulating microtubule dynamics, we measured dynamic astral microtubules over time in living cells (Fig 6D). We predicted that mutants lacking the acidic patch would exhibit microtubules that are more stable. Removing the β-CTT (tub2-430Δ) leads to significantly longer microtubules compared with that in wild-type controls (Fig 6E). Polymerization and depolymerization rates of these mutant microtubules were similar to those of wild-type controls; however, the catastrophe frequencies were significantly reduced (Fig 6F–H and Table 1). In contrast, mutants that retain the acidic patch (tub2-438Δ) exhibit microtubule lengths and dynamics that are similar to those of wild-type controls (Fig 6E–H

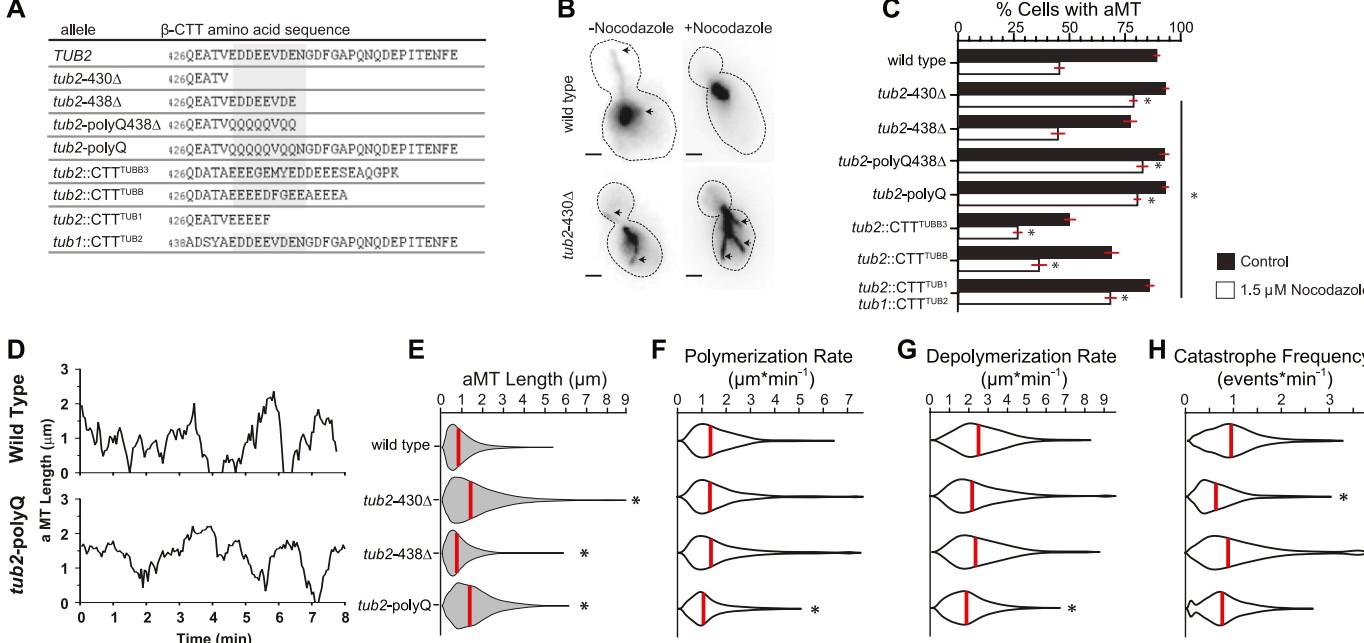

**Figure 6. β-CTT function requires negatively charged amino acids.**
**(A)** Amino acid sequences of β-CTTs of mutant yeast strains characterized in Fig 6. The native (TUB2) allele provided for comparison. **(B)** Representative images of preanaphase yeast strains expressing microtubules labeled with GFP-Tub1 treated with either 1.5 μM nocodazole or DMSO as a control. Arrowheads indicate the plus ends of astral microtubules (aMTs). **(C)** Percent of cells exhibiting aMTs after exposure to 1.5 μM nocodazole or DMSO for 1 h. Bars represent the pooled mean ± standard error of the proportion of four separate experiments pooled. Each group is composed of n ≥ 735 cells. **P « 0.001. Significance determined by Fisher's exact test. **(D)** Representative aMT life plots of wild-type and tub2-polyQ cells expressing GFP-labeled tubulin. Microtubule lengths were measured at 3- to 4-s time intervals. **(E)** Distributions of aMT lengths in living preanaphase cells. Images were collected at 3- to 4-s intervals for 10 min. Results represent pooled data from at least four separate experiments and at least 40 aMTs sampling n ≥ 3,593 time points for each strain. Red bars indicate median values. **P « 0.001. Significance determined by the Mann–Whitney U test. **(F)** Distributions of aMT polymerization rates, from cells analyzed in **(E)**. **(G)** Distributions of aMT depolymerization rates, from cells analyzed in **(E)**. **(H)** Distributions of aMT catastrophe frequency, from cells analyzed in **(E)**.

**Table 1.  Dynamics of astral microtubules measured in yeast cells during preanaphase.**

|  | Polymerization rate (μm/min) | Depolymerization rate (μm/min) | Polymerization duration (s) | Depolymerization duration (s) | Catastrophe frequency (events/min) | Rescue frequency (events/min) |
|---|---|---|---|---|---|---|
| Wild type | 1.59 (1.40–1.78) | 2.80 (2.52–3.07) | 40 (35–45) | 24 (21–27) | 1.00 (0.78–1.22) | 1.67 (1.31–2.02) |
| *tub2*-430Δ | 1.57 (1.34–1.80) | 2.49 (2.19–2.78) | 44* (38–50) | 36* (32–41) | 0.69* (0.52–0.85) | 0.79* (0.53–1.06) |
| *tub2*-438Δ | 1.61 (1.24–1.98) | 2.65 (2.30–3.00) | 40 (33–47) | 28 (24–32) | 0.94 (0.55–1.33) | 1.08* (0.69–1.48) |
| *tub2*-polyQ | 1.31*(1.15–1.47) | 2.21* (2.01–2.41) | 43 (37–49) | 35* (31–39) | 0.82 (0.67–0.97) | 0.91* (0.58–1.23) |

Source data are available for this table.

Image series were collected at 3- to 4-s intervals for 10 min and microtubule lengths were measured at each time point. Values shown are the medians, with 95% CIs in parentheses, of pooled data from at least four separate experiments.

*$P < 0.05$, compared with wild type, based on the Mann–Whitney $U$ test. Number of aMTs measured for each genotype: wild type, n = 45; *tub2*-430Δ, n = 51; *tub2*-438Δ, n = 34; and *tub2*-polyQ, n = 65.

and Table 1). Neutralizing negatively charged side chains in the acidic patch (*tub2*-polyQ) leads to longer microtubules with slower polymerization and depolymerization rates (Fig 6E–H and Table 1). Together, these results indicate that the negatively charged side chains in the acidic patch regulate the equilibrium between free heterodimers and microtubule polymers, and ablating this charged region shifts the equilibrium toward the polymer.

We next extended our analysis to ask whether sequence differences found in the tail regions of human β-tubulin isotypes might elicit different effects on microtubule stability. We tested this by generating chimeric β-tubulin composed of the yeast globular domain with the tail region replaced with that of either human class I or class III β-tubulins (Fig 6A). Both chimeras displayed sensitivity to nocodazole, with approximately half of cells losing astral microtubules, compared with untreated controls (Fig 6C). Interestingly, we note that the untreated controls for both chimeric mutants exhibit significantly lower percentages of cells with visible astral microtubules, compared with cells expressing wild-type yeast β-tubulin (Fig 6C). These results suggest that sequence differences between the yeast and human β-CTTs may lead to differential effects on microtubule dynamics.

Finally, we asked whether the role of β-CTT in regulating microtubule stability depended on its location on the tubulin heterodimer. We tested this by examining mutants that swapped the α- and β-CTT sequences, replacing the native yeast α-CTT with β-CTT sequence and vice versa (Fig 6A) (23). We found that swapping the tails results in an intermediate nocodazole sensitivity phenotype; a greater percentage of cells exhibited astral microtubules than wild-type controls, but not as great as that seen in *tub2*-430Δ (Fig 6C). This suggests that the function of β-CTT may be partly rescued by replacing it with the shorter α-CTT or by putting a longer tail on α-tubulin. Taken together, our findings indicate that the negatively charged β-CTT is a potent regulator of the tubulin equilibrium.

## Discussion

CTTs are the most molecularly diverse regions of tubulin proteins and have, therefore, generated great interest as potential sites for regulating microtubule function. In this study, we identify an important role for β-CTT in regulating microtubule dynamics. Our findings build on previous studies that discovered that the removal

of β-CTT by subtilisin enhances tubulin assembly in bulk assays (12, 13, 14). Our study provides the first insight into how β-CTT impacts assembly and dynamics at the level of individual microtubules by measuring microtubules growing from GMPCPP-stabilized seeds in vitro or from microtubule organizing centers in living cells. We directly demonstrate that β-CTT inhibits the microtubule polymer state by inhibiting tubulin assembly and promoting disassembly. Our results also reveal an unexpected role for β-CTT in regulating the structure and stability of microtubule plus ends.

The effect of β-CTT on inhibiting the microtubule polymer state may be explained by an electrostatic repulsion model in which the negatively charged β-CTT of one αβ-tubulin heterodimer repels the negatively charged CTTs of nearby heterodimers. The electrostatic repulsion model predicts that β-CTT could impact the microtubule polymer in two ways: first, by decreasing the affinity of heterodimers for the crowded and densely charged environment of the growing microtubule end and, second, by increasing the depolymerization rate by promoting the outward splaying of protofilaments at the shrinking microtubule end. In support of this model, we found that S-tubulin polymerizes faster and depolymerizes slower than undigested tubulin in vitro and exhibits a significantly higher affinity for microtubule polymer, as measured by the off-rate in the absence of free tubulin (Figs 1 and 2E). In addition, we found that budding yeast mutants that remove β-CTT or neutralize negatively charged amino acid side chains within β-CTT result in longer microtubules that are resistant to nocodazole (Fig 6). Together, our findings are consistent with an electrostatic repulsion model where the negative charge of β-CTT inhibits the microtubule polymer state and promotes the free heterodimer state.

In addition to regulating the equilibrium between free tubulin heterodimers and the microtubule state, our results reveal a surprising effect of β-CTT on the structure and stability of plus ends. Despite their faster polymerization, S-tubulin microtubules undergo catastrophe sooner than control microtubules in our dynamics experiments (Fig 1F) and washout experiments (Fig 3C and D). This suggests that β-CTT may normally stabilize the plus end and raises an apparent paradox—how could β-CTT stabilize the plus end while lowering the affinity of tubulin for the microtubule? We propose that β-CTT might regulate plus ends in two ways, which are relevant to the "GTP cap" and "end structure" models for dynamic instability. First, β-CTT may promote the enrichment or retention of GTP-tubulin at the plus end, leading to a larger GTP cap. We

speculate that this could arise either through a yet undefined role for β-CTT in regulating GTP hydrolysis once tubulin subunits assemble the microtubule lattice or through preventing the direct incorporation of GDP-tubulin into the plus end. Interestingly, our experiments in yeast, which are known to retain more GTP-tubulin in the microtubule lattice (28), do not show increased catastrophes when β-CTT is removed. Instead, we see a slight but significant decrease in catastrophe frequency when β-CTT is removed (Fig 6H). Whether this discrepancy can be attributed to differences between tubulins from different species, the activity of microtubule-associated proteins in vivo, or differences between buffer conditions and the conditions of the cytoplasm is an important question.

Second, the effect of β-CTT on subunit–polymer affinity might guide the structure of the growing plus end and/or prevent lattice defects. It is clear from previous studies that tubulin with β-CTT favors assembly into microtubule lattices, whereas S-tubulin can spontaneously nucleate nonmicrotubule aggregates or aberrant microtubule structures, including incomplete or "hooked" lattices (16). We also observed that after long periods (>1 h) in polymerization conditions, S-tubulin formed nonmicrotubule aggregates that were distinct from the microtubules extending from GMPCPP-stabilized seeds in our TIRF assays (data not shown). This indicates that β-CTT is important for inhibiting aberrant tubulin assembly. In addition, our TIRF assays reveal that microtubules assembled with S-tubulin tend to exhibit blunt plus ends, in contrast to the tapered plus ends seen in control microtubules (Fig 2). It is not obvious that these blunt ends represent a form of lattice defect; however, our current experiments do not permit the identification of incomplete lattices or changes in protofilament number. Alternatively, blunt ends could represent complete or nearly complete cylindrical lattices at the plus end. Because the blunt ends of S-tubulin microtubules are less stable after free tubulin washout, this raises the possibility that plus ends with complete lattices may have different allosteric coupling than plus ends with tapered ends with incomplete lattices. Testing these models will be the focus of future studies.

Our results also indicate that the role of β-CTT in regulating the microtubule dynamics may involve interactions with multivalent cations. Divalent cations are known to have important and divergent effects on microtubule structure and dynamics. Zinc and manganese ions promote the assembly of tubulin sheets and rings, respectively, rather than microtubule cylinders (29, 30). Calcium and magnesium ions potently destabilize microtubules (7, 31, 32). In contrast, oligocations, such as spermine, stabilize microtubules (33). Interestingly, α-tubulin CTT peptides bind to calcium ions and spermine (34). To our knowledge, β-CTT peptides have not been examined. Nevertheless, these results, together with our findings that magnesium ions require β-CTT to accelerate depolymerization, support a model in which multivalent cations could alter microtubule dynamics by binding and, perhaps, cross-linking negatively charged amino acid side chains in the CTTs. Such cross-linking could destabilize microtubules, as in the case of calcium and magnesium, by promoting the curling of protofilaments away from the lattice (35, 36, 37). Alternatively, cross-linking with oligocations could stabilize microtubules by strengthening the lattice, enhancing the local concentration of free tubulin subunits, or bundling microtubules (33, 38, 39).

Given its potent effect on microtubule assembly and structure, β-CTT is likely to be a key point for regulating microtubule dynamics

in cells. It is worth noting that our studies in yeast show that the effect of α-CTT on microtubule dynamics is minor, compared with that of β-CTT (23). The different effects of α-CTT versus β-CTT could be attributable to the shorter length of the yeast α-CTT, which contains fewer negatively charged side chains than the yeast β-CTT, or its position on the heterodimer (Fig 6A and C). Most cells in higher eukaryotes contain a complex blend of heterodimer species with CTTs that differ in genetically encoded β-CTT and α-CTT sequences and posttranslational modifications. Changing the blend of heterodimer species can profoundly alter microtubule dynamics in vivo (40), and this can be recapitulated in vitro using mixtures of recombinant tubulins (41, 42). Our findings indicate that differences in β-CTT composition could promote differences in assembly and plus-end stability across heterodimer species, providing a mechanism to tune microtubule dynamics in vivo.

# Materials and Methods

Chemicals and reagents were from Fisher Scientific and Sigma-Aldrich, unless stated otherwise.

### In vitro microtubule dynamics assays

Assays to measure microtubule dynamics by TIRF microscopy were based on previously described methods (43). Double-cycled microtubule seeds were assembled by incubating 20 μM rhodamine tubulin (Cytoskeleton, Inc.) in Britton–Robinson buffer 80 (BRB80) (80 mM PIPES brought to pH 6.9 with KOH, 1 mM EGTA, and 1 mM MgCl$_2$; minor pH adjustments were made with NaOH) with 1 mM GMPCPP at 37°C for 30 min. The sample was then centrifuged at 100,000 $g$ for 10 min at 30°C and the supernatant was removed. The pellet was suspended in 0.8× starting volume of ice-cold BRB80 buffer to depolymerize labile microtubules. An additional 1 mM GMPCPP was added and the microtubules were polymerized at 37°C for 30 min and pelleted again. The pellet was suspended in 0.8× starting volume of warm BRB80. The reaction was then gently pipetted 8–10 times to shear the microtubules, aliquoted into 1-μl volumes, and either used immediately or snap-frozen and stored at −80°C.

Imaging chambers were assembled using 22 × 22-mm and 18 × 18-mm coverslips. The coverslips were cleaned and silanized as previously described (43). The prepared glass coverslips were stored in desiccators at room temperature until used. The coverslips were mounted in a custom-fabricated stage insert and sealed with melted strips of parafilm. GMPCPP-stabilized microtubule seeds were affixed to coverslips using anti-rhodamine antibodies (Cat No.: A-6397; Fisher Scientific; diluted 1:50 in BRB80). The chambers were flushed with 1% Pluronic-F127 in BRB80 to prevent other proteins from adhering to the glass and equilibrated with an oxygen-scavenging buffer (40 mM glucose, 1 mM Trolox, 64 nM catalase, 250 nM glucose oxidase, and 10 mg/ml casein) before free tubulin addition. The imaging buffer consisting of unpolymerized tubulin (15–20% HiLyte-488–labeled tubulin [Cytoskeleton, Inc.] and 85–90% unlabeled porcine brain tubulin), 5 mM MgCl$_2$, 1 mM GTP, the oxygen-scavenging buffer, and BRB80 to 50 μl volume was then flowed into the imaging chambers. The chamber was sealed with VALAP (1:1:1 vaseline:lanolin:paraffin

wax) and warmed to 37°C using an ASI 400 Air Stream Stage Incubator (Nevtek) for 5 min before imaging. Temperature was verified using an infrared thermometer.

Images were collected on a Nikon Ti-E microscope equipped with a 1.49 NA 100× chrome-free infinity 160 (CFI160) Apochromat objective, TIRF illuminator, OBIS 488-nm and Sapphire 561-nm lasers (Coherent), and ORCA-Flash 4.0 LT scientific complementary metal–oxide–semiconductor camera (Hamamatsu Photonics) using Nikon Instruments Software (NIS) Elements software (Nikon). Images were acquired using two-channel, single-plane TIRF at 3-s intervals.

### Image analysis

Images were analyzed using a custom-made MATLAB program. Seeds were identified by thresholding image intensity and then the images were rotated and segmented along the axis of the microtubule. The images were then automatically cropped to four pixels above and below the microtubule axis and then max-projected into a single line of pixels for each time point. The time points were stacked to generate kymographs for analysis.

Polymerization and depolymerization rates were calculated by measuring the changes in microtubule length and time from the first and last points of the individual polymerization and depolymerization events from the kymographs. Polymerization rate constants were estimated by multiplying the slope of the polymerization rate linear model by the number of subunits in 1 μm and dividing by 60 to yield subunits per second (~1,750 subunits in a 14-protofilament microtubule nucleated by a GMPCPP seed). Depolymerization rate constants were determined similarly but by using the median depolymerization rate (μm·min$^{-1}$) from all tubulin concentrations pooled.

### Tubulin washout experiments

For washout experiments, GMPCPP-seeded imaging chambers were similarly assembled but not sealed with VALAP. The imaging chambers were warmed on the stage for 2–3 min, allowing the temperature to equilibrate to 37°C, and then dynamic microtubules were imaged for 30 s before free tubulin was removed from the imaging chamber with 4-5× chamber volumes of warm reaction buffer, without tubulin. Images were acquired continuously during the experiment at 1-s intervals.

The images were processed as described above, with the addition of postacquisition image stabilization that was used to reduce minor XY drift during image acquisition using the Image Stabilizer Plugin for ImageJ (44). Microtubule lengths were determined at each time point using intensity thresholding. We calculated instantaneous polymerization velocities using intensity thresholding to determine microtubule length at each time point. The microtubule lengths were fit to a linear regression to determine average polymerization rate at 10 s before washout. Washout initiation was defined computationally as the time point when background fluorescence was significantly lower than the previous three time points. We defined washout termination similarly, by comparing the background fluorescence between time points. Washout duration was defined as the time between initiation and termination. The average washout duration for all washout experiments was ~5 s.

Time to catastrophe was determined using a custom MATLAB program designed to iteratively fit groups of five microtubule lengths to a linear regression beginning from the washout. A catastrophe event was defined empirically as a loss of more than 150 nm·s$^{-1}$ (approximately two pixels). The slow depolymerization rate was determined by linear regression of the microtubule lengths from washout to catastrophe. The fast depolymerization rate was similarly determined, from the catastrophe time until the microtubule depolymerized back to the GMPCPP seed.

### Determining tubulin concentration

For each experiment, the tubulin concentration was determined by running a sample of the imaging buffer containing unpolymerized tubulin on a 10% Bis–Tris SDS–PAGE gel, followed by staining with Coomassie blue. The concentration was determined by averaging the intensities of triplicates of each sample and comparing to standard curve made from bovine serum albumin standards run on the same gel.

### Subtilisin treatment

S-tubulin was prepared by treating 10 μM porcine brain tubulin with 1% (w/w) Subtilisin (Cat No.: P5380; Sigma-Aldrich) at 30°C for 5 min. Proteolysis was inactivated with 1 mM PMSF and samples were moved to ice for 20 min to depolymerize microtubules. The S-tubulin was concentrated in Amicon Ultra centrifugal concentrator columns (Cat No.: UFC501096; EMD Millipore) and spun at 5,000 g for 3 min at 4°C. Between spins, the concentrated S-tubulin was pipetted and visually checked for aggregation. At first signs of persistent aggregation, the concentrator tubes were placed on ice for 20 min and checked for aggregation again. When the samples were concentrated approximately 5× and no aggregation was visible, the S-tubulin was aliquoted, snap-frozen, and stored at −80°C. Stock concentration was determined as described above.

### Proteomics analysis

Proteomic analyses of subtilisin-digested tubulin samples were performed using in-gel digestion of bands cut from an SDS–PAGE gel (45). The bands were reduced, alkylated, trypsin-digested, and then analyzed by nano-ultra-high performance liquid chromatography tandem mass spectrometry (UHPLC-MS/MS) using a nano-Easy II nano-LC and Thermo Q Exactive HF Orbitrap MS as described in reference 45. Peptides were separated on a self-made 20-cm C18 analytical column (100 μm) packed with 2.7 μm Phenomenex Cortecs C18 resin. After equilibration with 3 μl of 5% acetonitrile and 0.1% formic acid, the peptides were separated by a 70-min linear gradient from 4 to 30% acetonitrile with 0.1% formic acid at 350 nl/min. The mass spectrometer was operated in positive ion mode with data-dependent (top 15) acquisition. Full MS scans were obtained with a range of $m/z$ 300 to 1,800, 60,000 resolution, ions were fragmented using higher-energy collisional dissociation (28 NCE), and tandem mass spectra were acquired at a mass resolution of 15,000. The dynamic exclusion time was 20 s. Raw files were processed using Proteome Discoverer 2.2 and searched against the *Sus scrofa* (TUBA1B: Q2XVP4 and TUBB2B: A0A287AGU7) UniProt Knowledgebase and International Protein Index databases (release date: February, 2018)

using Mascot. Mass tolerances were ±10 ppm for MS parent ions and ±15 ppm for MS/MS fragment ions. Semi-trypsin specificity was used allowing for 1 missed cleavage. Variable modifications included Met oxidation, protein N-terminal acetylation, peptide N-terminal pyroglutamic acid formation, and cysteine carbamidomethylation. Secondary error-tolerant searches were performed to identify additional posttranslational modifications. Area under the curve was calculated and used for quantification at the peptide level.

## Western blotting

Samples were run on 10% Bis–Tris SDS–PAGE gels and transferred to polyvinylidene difluoride membranes. The membranes were blocked for 1 h in Odyssey blocking buffer (Cat No.: 927-40000; LI-COR Biosciences). All antibodies were diluted in Odyssey blocking buffer. The blots were probed for 60 min at room temperature using the following primary antibodies: pan α-tubulin (4A1) (46) at 1:100 and DM1A α-tubulin (T6199; Sigma-Aldrich) at 1:1,000; pan β-tubulin antibody (9F3; Cell Signaling Technology) at 1:1,000 against the N terminus of β-tubulin. Primary antibodies against tyrosinated-α-tubulin (T9028; Sigma-Aldrich) were used at 1:1,000. The blots were washed with PBS-Tween (PBS components and 0.1% Tween-20) before incubation with secondary antibodies, IRDye 800CW goat anti-rabbit (P/N: 926-32211; LI-COR Biosciences) and IRDye 680RD goat anti-mouse (P/N 926-68070; LI-COR Biosciences) antibodies at 1:15,000 in blocking buffer, for 60 min at room temperature in the dark. The blots were washed three times with PBS-T and then again with PBS before being imaged on the Odyssey Imaging System using the Image Studio software (LI-COR Biosciences). Band intensities were measured using ImageJ after export from the Image Studio software.

## Yeast strains and manipulation

General yeast manipulation and media and transformation were performed by standard methods (47). A detailed list of strains is provided in Table S2. GFP-*TUB1* fusion was integrated and expressed from the *LEU2* chromosomal locus, so that the fusion protein was expressed in addition to the native *TUB1* (48, 49). Mutations to the β-tubulin tail were made at the native chromosomal locus as previously described (18, 23). Human β-tubulin tail chimeras were constructed by integrating the *TRP1* auxotrophic marker 331 base pairs downstream of the *TUB2* STOP codon. The genomic DNA from this strain was used as a template for PCR with PAGE-purified chimeric forward primers to amplify the 3′ end of the *TUB2* coding sequence through the integrated *TRP1* marker and replace the native yeast tail sequence after codon 427 with the human tail sequences. The PCR product was then transformed into naive competent yeast as previously described (47). Integration into transformed colonies was confirmed by sequencing the native *TUB2* locus. The *mnr2Δ* deletion mutant was generated by conventional methods (50).

## Growth assays measuring magnesium sensitivity

Cells were grown to saturation in synthetic complete media containing 4 mM MgSO$_4$ (Sunrise Science Products) and then pelleted and washed twice with synthetic media lacking MgSO$_4$. After washing, the cells were diluted 1:50 into 96-well plates with 200 μl/well of

either synthetic complete media containing 4 mM MgSO$_4$ or synthetic complete media lacking MgSO$_4$, but supplemented with MgCl$_2$. OD600 values were measured at 5-min intervals over 21 h at 30°C with orbital shaking using a Cytation 3 plate reader (Biotek). Doubling time was calculated by fitting the OD600 values, recorded over time, to an exponential curve using a custom MATLAB code as described previously (18). Normalized doubling times are based on data from at least four independent isolates for each genotype, measured in five separate experiments.

## Measuring microtubule length and dynamics in yeast

Assays were performed as described previously (26). Cells were grown overnight in a shaking incubator at 30°C in synthetic complete media, and then diluted into fresh media, and returned to the shaking incubator for ~4 h before imaging. For experiments at low Mg$^{2+}$, the cells were diluted into synthetic complete media lacking MgSO$_4$. Before imaging, the cells were gently pelleted and mounted in parafilm slide chambers were made and coated with concanavalin A (2 mg/ml; Cat No.: C5275; Sigma-Aldrich) as described previously (26).

Images were collected on a Nikon Ti-E microscope equipped with a 1.45 NA 100× CFI Plan Apo objective, piezo electric stage (Physik Instrumente), spinning disk confocal scanner unit (CSU10; Yokogawa), 488-nm and 561-nm lasers (Agilent Technologies), and an EMCCD camera (iXon Ultra 897; Andor Technology) using NIS Elements software (Nikon). During acquisition, the temperature of the stage was 25°C. For experiments in G1 cells, microtubule lengths were measured in maximum-intensity projections of Z series consisting of 29 planes separated by 300 nm, collected at single time points. Individual microtubules were distinguished from microtubule bundles based on the signal intensity of GFP-labeled α-tubulin. For experiments measuring dynamic microtubules in preanaphase cells, Z series consisting of 13 images separated by 500 nm were collected at 5-s intervals.

Polymerization and depolymerization events were defined as at least three contiguous data points that produced a length change ≥0.5 μm with a coefficient of determination ≥0.80. Polymerization and depolymerization rates of individual events were determined by dividing the change in microtubule length by the change in seconds, and multiplied by 60 to convert seconds to minutes. Rates are reported in μm·min$^{-1}$. Catastrophe frequencies were determined for individual astral microtubules by dividing the number of catastrophe events by the total lifetime of the microtubule, minus time spent in disassembly. Rescue frequencies were determined for individual astral microtubules by dividing the number of rescue events by the total lifetime, minus time spent in assembly. At least 34 astral microtubules were analyzed for each genotype. Dynamics measurements for individual microtubules were pooled for the genotype and then compared with pooled data for different genotypes using the Mann–Whitney U test to assess whether the values for different data sets are significantly different.

## Nocodazole sensitivity assay

Cells were grown overnight in rich liquid media in a shaking incubator at 30°C, diluted, and grown to early log phase in fresh media the following day. The cultures were brought to 1.5% DMSO and nocodazole was added to 1.5 μM. The cells were then returned to the 30°C shaking incubator for 60 min. Fixative (18.5% formaldehyde

and 0.5 M KPO$_4$) was added to the cultures at a ratio of 1:3 and the cultures were returned to the shaker at 30°C for 3 min. The cultures were pelleted at 1,500 $g$ for 2 min, the supernatant was removed, and the cells were suspended in a quencher solution (0.1% Triton-X, 0.1 M KPO$_4$, and 10 mM ethanolamine). The cells were pelleted, supernatant decanted, and washed twice in 0.1 M KPO$_4$. The fixed cells were loaded into the coated slide chambers and washed with 0.1 M KPO$_4$, and the chambers were sealed with VALAP.

Images were collected on a Nikon Ti-E wide-field microscope equipped with a 1.49 NA 100× CFI160 Apochromat objective and an ORCA-Flash 4.0 LT scientific complementary metal–oxide–semiconductor camera (Hamamatsu Photonics) using NIS Elements software (Nikon). Images were acquired using 100-ms exposure, 11 $z$-planes, with 300 nm separation.

The cells were segmented using a custom ImageJ macro previously described ([18]). The data were blinded for analysis, and the segmented cells were visually scored for the presence of astral microtubules using a custom MATLAB code.

# Supplementary Information

# Acknowledgements

We thank Dr. Melissa Gardner (University of Minnesota) for technical assistance in developing in vitro assays to measure microtubule dynamics, Dr. Jay Gatlin (University of Wyoming) for helping with the purification of porcine brain tubulin, Dr. Michael McMurray (University of Colorado School of Medicine) for sharing equipment, and members of the Moore laboratory for their helpful discussions and advice. We are grateful to Drs. Kirk Hansen and Monika Dzieciatkowska (University of Colorado School of Medicine) for conducting the proteomic analysis and for guidance in interpreting the data. This work was supported by National Institutes of Health grant R01GM112893 (to JK Moore).

## Author Contributions

C. Fees: conceptualization, data curation, formal analysis, validation, investigation, methodology, resources, software, visualization, and writing—original draft, reviewing, and editing.
J. Moore: conceptualization, resources, data curation, formal analysis, supervision, funding acquisition, validation, investigation, visualization, methodology, and writing—original draft, project administration, review, and editing.

## Conflict of Interest Statement

The authors declare that they have no conflict of interest.

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
