## [Reviewer comments · Life Science Alliance]

Regulation of microtubule dynamic instability by the carboxy-terminal tail of β -tubulin

Colby P. Fees and Jeffrey K. Moore

DOI: 10.26508/lisa.201800054

Review timeline:	Submission Date:	21 March 2018
	Revision Received:	21 March 2018
	Editorial Decision:	21 March 2018
	Accepted:	26 March 2018

Report:

(Note: Letters and reports are not edited. The original formatting of letters and referee reports may not be reflected in this compilation.)

Please note that the manuscript was previously reviewed at another journal and the reports were taken into account in the decision-making process at Life Science Alliance. Since the original reviews are not subject to Life Science Alliance's transparent review process policy, the reports and author response cannot be published.

Please note that the manuscript was previously reviewed at another journal and the reports were taken into account in inviting a revision for publication at *Life Science Alliance* prior to submission to *Life Science Alliance*.

1st Editorial Decision

21 March 2018

Thank you for submitting your revised manuscript entitled "Regulation of microtubule dynamic instability by the carboxy-terminal tail of β -tubulin". Your manuscript was reviewed at another journal before, and the referee reports of this previous round of review were confidentially transferred to us with your permission.

The reviewers who evaluated your work had noted that your work is robust, but that the manuscript could benefit from better explanations, including explanation of why you tested effects of magnesium on microtubule polymerization / depolymerization. They furthermore noted that a better analysis of microtubule dynamics in vivo would be beneficial, and that there are critical differences between yeast and porcine tubulin that should be taken into account, ideally including in vitro analysis of yeast microtubule dynamics.

You have provided a revised manuscript addressing all these points except for the in vitro analysis of yeast microtubule dynamics. You added further in vivo analysis of microtubule dynamics as well as analyses of how changing the concentration of magnesium ions impacts polymerization and depolymerization of S-tubulin and untreated controls. We evaluated this revised version of your manuscript and appreciate the introduced changes. We would therefore be happy to publish your paper in Life Science Alliance pending final revisions necessary to meet our formatting guidelines. I list below a few items you should pay attention to allow production of your manuscript.

Congratulations on this very nice work!